# Diversity Analysis of Tick-Borne Viruses from Hedgehogs and Hares in Qingdao, China

Geng Hu,[a] Fachun Jiang,[b] Qin Luo,[a] Kexin Zong,[a] Liyan Dong,[b] Guoyong Mei,[a] Haijun Du,[a] Hongming Dong,[a] Qinqin Song,[a] Juan Song,[a] Zhiqiang Xia,[a] Chen Gao,[a] Jun Han[a]

aState Key Laboratory of Infectious Disease Prevention and Control, National Institute for Viral Disease Control and Prevention, Chinese Center for Disease Control and Prevention, Beijing, China
bQingdao Municipal Center for Disease Control and Prevention, Qingdao Institute of Prevention Medicine, Qingdao, Shandong Province, China

Geng Hu and Fachun Jiang contributed equally to this work. The order of names for the two firstco-authors was determined arbitrarily.

**ABSTRACT** Tick-borne viruses (TBVs) have attracted increasingly global public health attention. In this study, the viral compositions of five tick species, *Haemaphysalis flava*, *Rhipicephalus sanguineus*, *Dermacentor sinicus*, *Haemaphysalis longicornis*, and *Haemaphysalis campanulata*, from hedgehogs and hares in Qingdao, China, were profiled via metagenomic sequencing. Thirty-six strains of 10 RNA viruses belonging to 4 viral families, including 3 viruses of *Iflaviridae*, 4 viruses of *Phenuiviridae*, 2 viruses of *Nairoviridae*, and 1 virus of *Chuviridae*, were identified in five tick species. Three novel viruses of two families, namely, Qingdao tick iflavirus (QDTIFV) of the family of *Iflaviridae* and Qingdao tick phlebovirus (QDTPV) and Qingdao tick uukuvirus (QDTUV) of the family of *Phenuiviridae*, were found in this study. This study shows that ticks from hares and hedgehogs in Qingdao harbored diverse viruses, including some that can cause emerging infectious diseases, such as Dabie bandavirus. Phylogenetic analysis revealed that these tick-borne viruses were genetically related to viral strains isolated previously in Japan. These findings shed new light on the cross-sea transmission of tick-borne viruses between China and Japan.

**IMPORTANCE** Thirty-six strains of 10 RNA viruses belonging to 4 viral families, including 3 viruses of *Iflaviridae*, 4 viruses of *Phenuiviridae*, 2 viruses of *Nairoviridae*, and 1 virus of *Chuviridae*, were identified from five tick species in Qingdao, China. A diversity of tick-borne viruses from hares and hedgehogs in Qingdao was found in this study. Phylogenetic analysis showed that most of these TBVs were genetically related to Japanese strains. These findings indicate the possibility of the cross-sea transmission of TBVs between China and Japan.

**KEYWORDS** tick, tick-borne virus, virus diversity, next-generation sequencing, evolutionary analysis

Address correspondence to Jun Han, hanjun_sci@163.com.

The authors declare no conflict of interest.

At present, more than 100 pathogens have been found in ticks, including bacteria, fungi, protozoa, and viruses. For humans, ticks are the second most important arboviral vector after mosquitoes, and most of the pathogens causing serious harm to humans are tick-borne viruses (TBVs). The earliest known tick-borne pathogenic virus was discovered more than 100 years ago, namely, Loupingill virus (1), which is the pathogen causing severe encephalitis in sheep and other livestock. Since then, a growing number of TBVs have been found, many of which are zoonotic pathogens (2). Some even cause highly infectious or high-mortality-rate epidemics, posing a threat to animal and human health. In recent years, many human diseases caused by emerging tick-borne viruses have been reported, such as Dabie bandavirus (severe fever with thrombocytopenia syndrome virus [SFTSV]) (3), Jingmen tick virus (JMTV) (4, 5), Alongshan virus (ALSV)

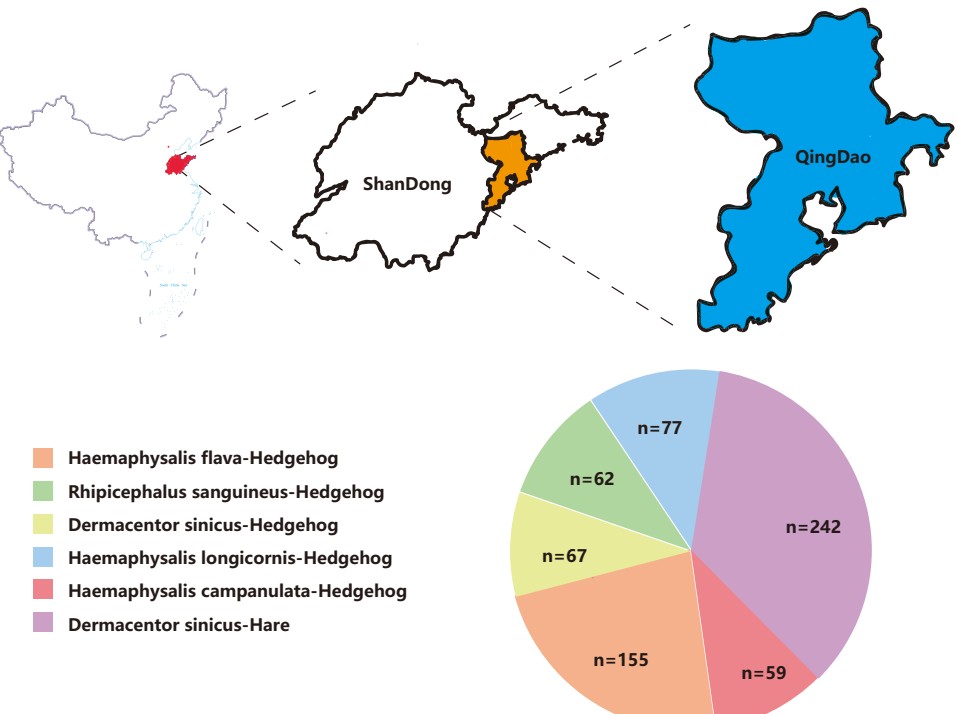

**FIG 1** Location of Qingdao in China (top) and composition of tick species (bottom). (Top) Qingdao is located in northeast Shandong, China. Qingdao was enlarged from the map of China. (Bottom) The types of ticks are represented by different colors. Orange, *Haemaphysalis flava* fed on hedgehogs; green, *Rhipicephalus sanguineus* fed on hedgehogs; yellow, *Dermacentor sinicus* fed on hedgehogs; blue, *Haemaphysalis longicornis* fed on hedgehogs; red, *Haemaphysalis campanulata* fed on hedgehogs; purple, *Dermacentor sinicus* fed on hares. The China map was downloaded from http://bzdt.ch.mnr.gov.cn/, and the map of Shandong Province was downloaded from the Shandong Province marking map service website (http://bzdt.ch.mnr.gov.cn/browse.html?picId=%224o28b0625501ad13015501ad2bfc0210%22) and edited using Adobe Illustrator.

(6), Songling virus (SGLV) (7), Beiji nairovirus (8), and Langya henipavirus (9). This indicates that continuous surveillance of viruses carried by ticks is necessary.

In addition, with the development of next-generation sequencing (NGS), an increasing number of tick-borne viruses have been gradually discovered. NGS has become a powerful tool for discovering arboviruses from blood-sucking arthropod vectors, which can help us better understand the virus population and discover unknown viruses. In fact, many known or potential tick-borne viruses have been found through viral genome analysis by NGS (10–16). It is worth noting that research based on NGS shows that the viruses in ticks have unprecedented diversity. At present, new virus species have been identified, including *Astroviridae*, *Reoviridae*, *Flaviviridae*, *Bunyaviridae*, *Paramyxoviridae*, and *Nanoviridae* (11, 17).

China has a vast territory, spanning over 50 latitudes from north to south, with 6 climatic zones. There are different species of ticks and tick-borne viruses in each climatic zone. Currently, there have been a number of studies on tick-borne viruses in different Chinese provinces (10, 18). Qingdao, a coastal city in Shandong, China, faces South Korea and Japan across the sea (Fig. 1). Due to its location in the north temperate monsoon region and its marine climate, Qingdao was selected as the area to carry out research on tick-borne viruses. In this study, 662 ticks of 5 species were collected from hares and hedgehogs in Qingdao, Shandong, China. The viruses were identified by NGS and confirmed by reverse transcription-quantitative PCR (RT-qPCR). The diversity and evolution of tick-borne viruses were also analyzed and compared with those of viruses of the same family and genus.

## RESULTS

**Diversity of tick viruses.** A total of 662 ticks were collected from hares and hedgehogs in the suburbs of Qingdao, Shandong Province, from June to July 2019. Among

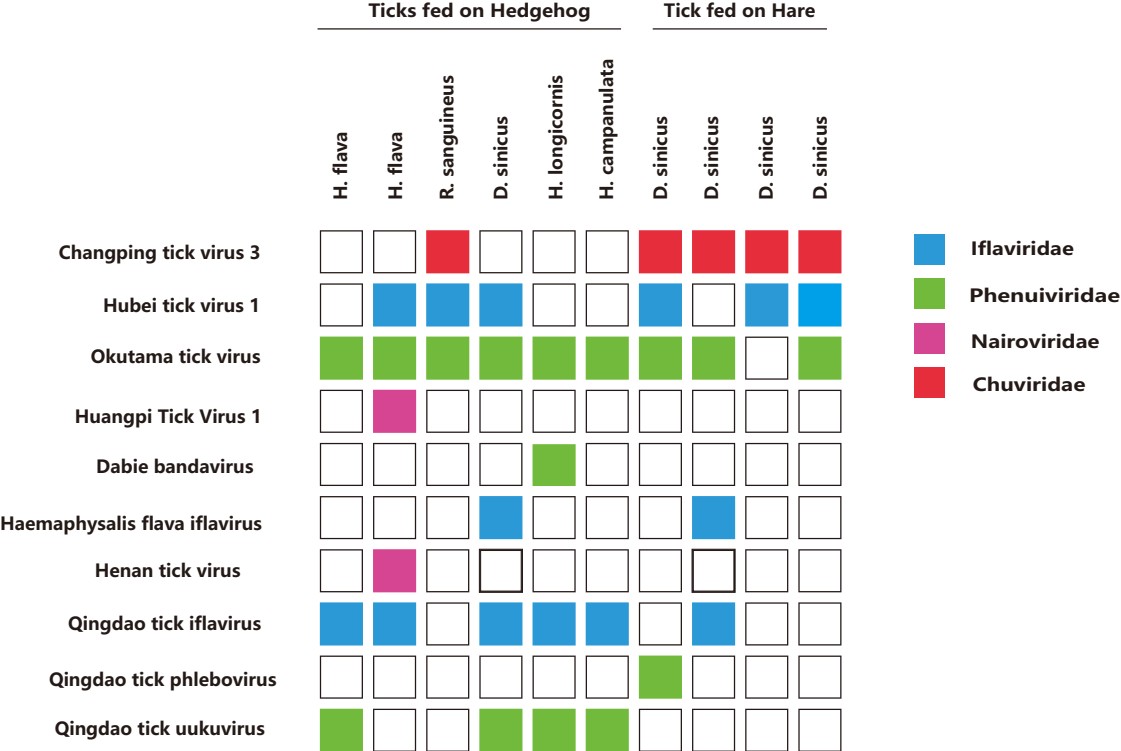

**FIG 2** List of viruses detected within tick samples.

them, ticks from hedgehogs were classified as follows: *Haemaphysalis flava* (*n* = 155), *Rhipicephalus sanguineus* (*n* = 62), *Dermacentor sinicus* (*n* = 67), *Haemaphysalis longicornis* (*n* = 77), and *Haemaphysalis campanulata* (*n* = 59). Ticks from hares belonged to only one species, *Dermacentor sinicus* (*n* = 242) (Fig. 1). The ticks were used for RNA library construction and sequencing. After quality control and adapter trimming, a total of 853,238,376 paired-end (PE) clean reads were generated (see Table S1 in the supplemental material), including 873,154 viral reads, accounting for 0.12% of the total RNA reads. After BLAST comparison and filtration of the virus host database, these virus groups were finally annotated and assigned to 10 viruses of 4 families, *Iflaviridae*, with a positive-sense single-stranded RNA [(+)ssRNA] genome, and *Phenuiviridae*, *Nairoviridae*, and *Chuviridae*, with negative-sense single-stranded RNA [(−)ssRNA] genomes (Fig. 2).

**Characteristics of viral genome structure and phylogeny.** Subsequently, the complete genomes of 36 strains from 10 representative viruses were successfully obtained by metagenomic bioinformatics analyses. Seven known viruses were found, Dabie bandavirus (SFTSV), Okutama tick virus (OKTV), Huangpi tick virus 1 (HPTV1), Changping tick virus 3 (CPTV3), Henan tick virus (HNTV), Haemaphysalis flava iflavirus (HfIFV), and Hubei tick virus 1 (HBTV1). Three new viruses were discovered, Qingdao tick iflavirus (QDTIFV), Qingdao tick phlebovirus (QDTPV), and Qingdao tick uukuvirus (QDTUV) (Fig. 2). These three viruses were named according to the viral species, the place of discovery (sample collection), and the closest relationships with previously described tick-associated viruses.

**Viral diversity and evolution. (i) (+)ssRNA viruses: HfIFV, HBTV1, and QDTIFV of *Iflaviridae*.** Fourteen strains of three genera, HfIFV, HBTV1, and a novel virus (QDTIFV), of *Iflaviridae* were found in ticks from Qingdao. Iflaviruses possess single-stranded, positive-sense, nonsegmented RNA genomes (19) with a single open reading frame (ORF) (20). The ORF is directly translated into a large polyprotein, which is subsequently processed into structural proteins, namely, capsid protein (N-terminal region),

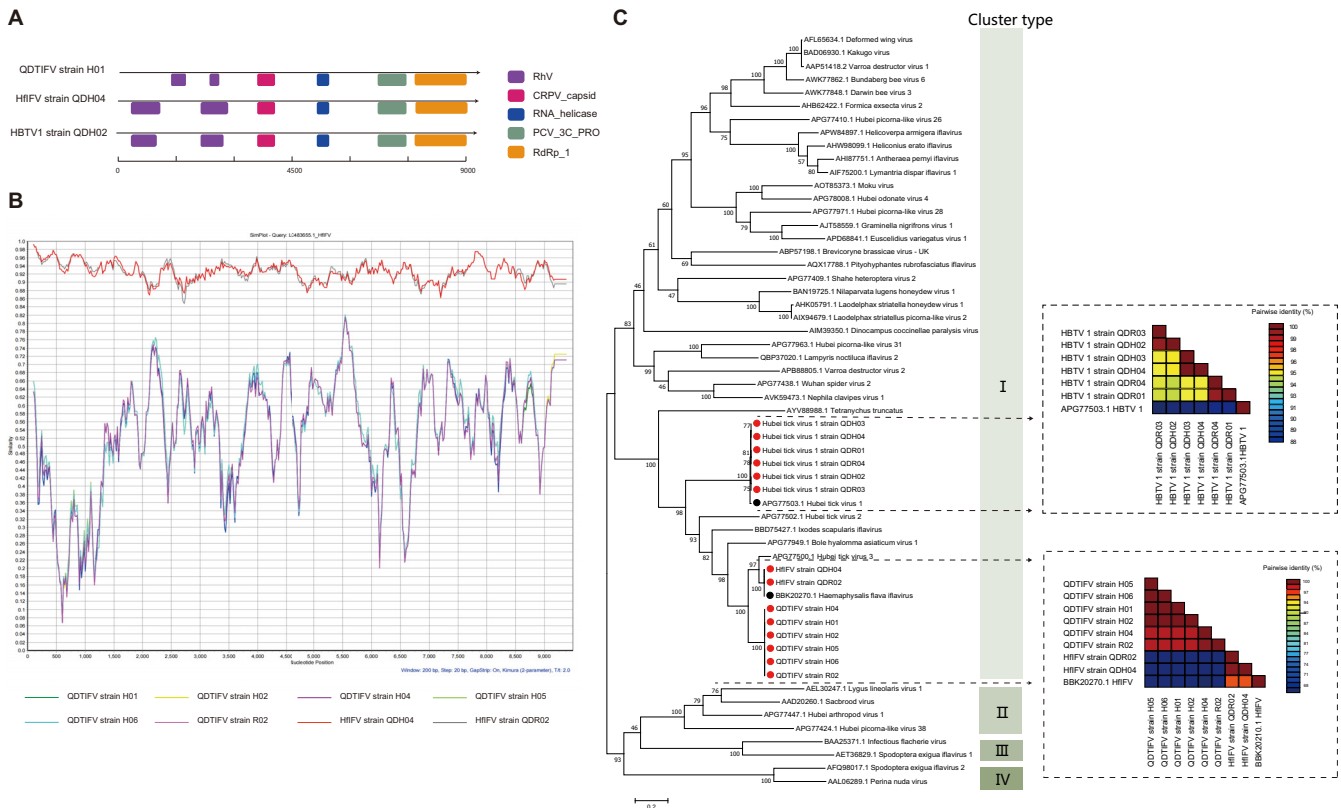

**FIG 3** Genome structure and phylogenetic analysis of iflaviruses. (A) Genome structure and putative ORFs of QDTIFV H01, HfIFV QDH04, and HBTV1 QD-H02. The viral genome contains one segment encoding RdRp, Picornavirus capsid protein (Rhv), cricket paralysis virus capsid protein like, RNA helicase, and Picornarin 3C-like protease (PCV_3C_PRO). The encoded proteins are marked by differently colored modules. All proteins are drawn to scale for genomic size. (B) Analysis of the genomic similarity of iflavirus strains with Haemaphysalis flava iflavirus strain 17TYM-T1 (GenBank accession number LC483655.1). The sequence of strain 17TYM-T1 was used as the query sequence. (C) Phylogenetic tree of iflaviruses based on the amino acid sequence of the RdRp. The heatmap of the paired genetic distances is based on the full-length nucleotide sequences. Viruses obtained in ticks here are highlighted in red. The bar indicates nucleotide substitutions per site. The GenBank accession numbers of the viruses here are detailed in Table S4 in the supplemental material.

and nonstructural proteins, namely, RNA-dependent RNA polymerase (RdRp) (C-terminal region) (Fig. 3A).

The HfIFV QD and QDTIFV strains had the closest evolutionary relationship to HfIFV (GenBank accession number LC483655.1), which was detected in Japan (15) (Fig. 3B). HfIFV strains QDH04 and QDR02 showed 93.08% to 93.12% nucleotide identity and 99.34% to 99.59% amino acid identity with HfIFV. The newly discovered QDTIFV strains (H01, H02, H04 to H06, and R02) fell within the iflavirus group, having the closest, yet distant, relationship (nucleotide identity of 66.4% to 66.6% and amino acid identity of 64.2% to 66.3%) to HfIFV, and it showed 65.4% to 65.6% nucleotide identity and 64.4% to 64.5% amino acid identity to Hubei tick virus 3 (HBTV3) (GenBank accession number KX883728.1) reported in China (17) (Table 1; Fig. S1). QDTIFV was defined as a novel virus according to the rule that the similarity with known virus sequences is less than 80%.

Phylogenetic trees were constructed using 57 iflavirus strains, including 14 strains from this study. Similarly, iflaviruses were classified into four clades based on their full-length sequences (21). Fourteen strains, including two HfIFV strains (QDH04 and QDR02), six HBTV1 strains (QDH02 to QDH04, QDR01, QDR03, and QDR04), and six QDTIFV strains (H01, H02, H04 to H06, and R02), were all located in cluster I. The pairwise similarity analysis based on the full sequences showed up to 94% nucleotide identity between the currently sequenced HBTV1 strains (QDH02 to QDH04, QDR01, QDR03, and QDR04) (Fig. 3C). Phylogenetically, the six QDTIFV strains (H01, H02, H04 to H06, and R02) and two HfIFV strains (QDH04 and QDR02) were clustered on a branch with HfIFV (GenBank accession number BBK20270.1) (15). The six QDTIFV strains (H01, H02, H04 to H06, and R02) were

**TABLE 1** Nucleotide and amino acid sequence identities of HfIFV strain QDH04 and Qingdao tick iflavirus strain H01 with other iflaviruses[a]

| | % identity | | | | | | | |
|---|---|---|---|---|---|---|---|---|
| | **HfIFV QDH04** | | | | **Qingdao tick iflavirus H01** | | | |
| | **Nucleotide** | | **Amino acid** | | **Nucleotide** | | **Amino acid** | |
| **Cluster and virus** | **Full length** | **RdRp** | **Full length** | **RdRp** | **Full length** | **RdRp** | **Full length** | **RdRp** |
| I | | | | | | | | |
| Haemaphysalis flava iflavirus | 92.5 | 93.3 | 99 | 99.7 | 66.5 | 71.1 | 64.2 | 75.1 |
| Hubei tick virus 1 | 52.1 | 63.6 | 34.2 | 62.0 | 51.6 | 65.2 | 34.2 | 62.0 |
| Hubei tick virus 3 | 74 | 77.5 | 79.8 | 87.1 | 65.4 | 71.2 | 64.4 | 76.6 |
| Varroa destructor virus 1 | 40.0 | 45.0 | 22.8 | 37.8 | 39.7 | 47.7 | 21.6 | 39.3 |
| II | | | | | | | | |
| Sacbrood virus | 41.6 | 51.5 | 20.8 | 62.0 | 41.1 | 50.6 | 21.0 | 62.0 |
| III | | | | | | | | |
| Infectious flacherie virus | 37.1 | 38.4 | 16.2 | 19.7 | 37.2 | 38.6 | 15.7 | 18.8 |
| IV | | | | | | | | |
| Perina nuda virus | 37 | 45.1 | 16.2 | 19.7 | 35.7 | 43.4 | 16.6 | 18.8 |

[a]The GenBank accession numbers of the viruses listed here are detailed in Tables S4 and S5 in the supplemental material.

located in an independent subgroup (Fig. 3C), and pairwise similarity analysis based on the full sequences showed up to 97% nucleotide identity among the six QDTIFV strains (H01, H02, H04 to H06, and R02) (Fig. 3C).

**(ii) (−)ssRNA viruses. (a) SFTSV, OKTV, QDTPV, and QDTUV of Phenuiviridae.** Fifteen virus strains from three viral genera of *Phenuiviridae* were found in ticks from the Qingdao area in this study. The genome of *Phenuiviridae* is divided into three segments, encoding at least four structural proteins (Fig. 4A), including the Length (L), Middle (M), and Small (S) segments (22). QDTPV was clustered with Mukawa virus (MUKV), a virus detected from *Ixodes persulcatus* ticks sampled in Japan in 2013, and it showed nucleotide identities of 75.8%, 68.3%, and 34.6% with the L, M, and S segments of Mukawa virus (GenBank accession numbers NC_043509.1 to NC_043511.1) and amino acid identities of 88.9%, 71.1%, 83.8%, and 57.3% with RdRp, G, N, and Ns of Mukawa virus (GenBank accession numbers YP_009666331 to YP_009666334), respectively (Table 2; Fig. S2 to S4). QDTUV (H01 and H04 to H06) was clustered with Toyo virus reported in Japan in 2021 (23), showing nucleotide identities of 77.88% to 78.29%, 72.48% to 78.70%, and 45.79% to 46.09% with the L, M, and S segments of Toyo virus (GenBank accession numbers LC618931.1 to LC618933.1), respectively, and amino acid identities of 90.05% to 91.29%, 81.04% to 81.48%, 85.14% to 86.28%, and 72.68% to 73.0% with RdRp, G, N, and Ns (GenBank accession numbers BCT55140.1 to BCT55143.1), respectively (Table 3; Fig. S5 to S7).

Phylogenetic analyses showed that two novel viruses, QDTPV and QDTUV, belong to the family *Phenuiviridae* (Fig. 4F). In this study, Dabie bandavirus strain QDH05 was clustered with strains isolated previously from Shandong, China (Fig. 4B); it had the closest relationship with a strain of Dabie bandavirus (GenBank accession number KR706567.1) that was isolated from a patient's serum (99.8% nucleotide identity) in China. From the phylogenetic tree, QDTPV was evolutionarily located on a previous branch of MUKV strain HLJ (GenBank accession number YP_009666332.1) and Kuriyama virus (GenBank accession number UXL90891.1) (13, 24) between tick-borne viruses and sandfly/mosquito-borne viruses (Fig. 4C). QDTUV, which had the closest genetic relationship with Toyo virus (13), was assigned to a subgroup of the Kaisodi group (Fig. 4D) with Silverwater virus (GenBank accession number YP_010086157.1), Huangpi tick virus 2 (GenBank accession number YP_009293590.1) (17), and Kaisodi virus (GenBank accession number AWW17495.1) on the phylogenetic tree, and pairwise similarity analysis based on the full sequences showed up to 90% nucleotide acid identity among four QDTUV strains (H01 and H04 to H06) (Fig. 4F). The M-segment-deficient phleboviruses

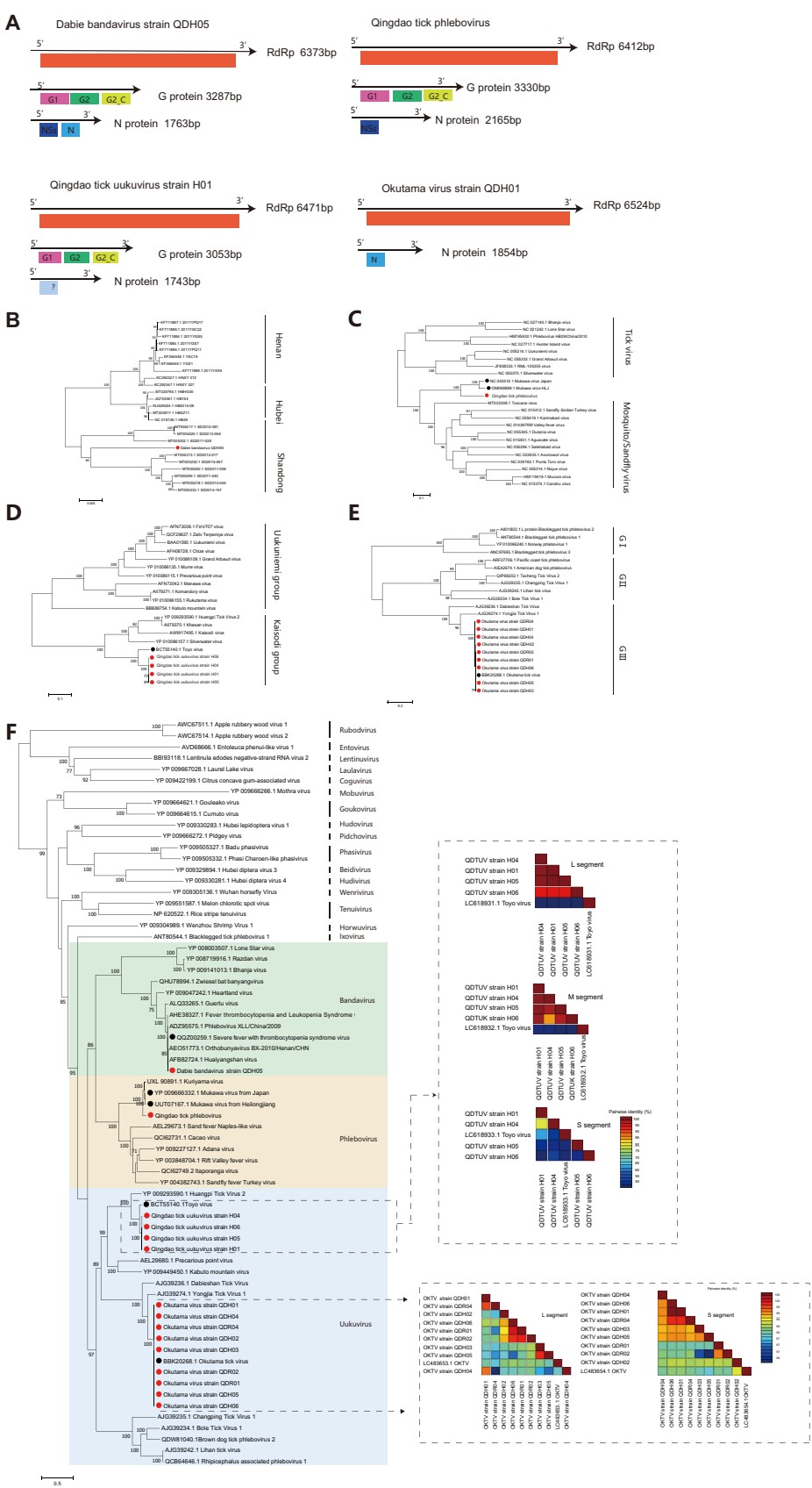

**FIG 4** Genome structure and phylogenetic analysis of *Phenuiviridae*. (A) Genome structure and putative coding regions of Dabie bandavirus, Qingdao tick phlebovirus, Qingdao tick uukuvirus strain H01, and

**TABLE 2** Nucleotide and amino acid sequence identities of QDTPV with other *Phenuiviridae*[a]

| | % identity with QDTPV | | | | | | |
| | Nucleotide | | | Amino acid | | | |
| Cluster type and virus | L | M | S | RdRp | G | N | Ns |
|---|---|---|---|---|---|---|---|
| Phlebovirus | | | | | | | |
| Mukawa virus | 75.8 | 68.3 | 34.6 | 88.9 | 71.1 | 83.8 | 57.3 |
| Kuriyama virus | 76.8 | 69.2 | 69.0 | 88.4 | 70.7 | 81.0 | 57.3 |
| Rift Valley fever virus | 54.4 | 43.5 | 47.5 | 49.0 | | 48.2 | 18.3 |
| Toscana virus | 53.5 | 41.2 | 31.1 | 47.7 | 27.7 | 45.7 | 21.4 |
| Sandfly fever Turkey virus | 53.7 | 42.5 | 31.4 | 48.1 | 30.6 | 46.1 | 16.5 |
| Bandavirus | | | | | | | |
| SFTSV | 45.8 | 36.1 | 41.7 | 31.8 | 19.4 | 36.6 | 18.5 |
| Uukuvirus | | | | | | | |
| Kabuto virus | 46.9 | 40.1 | 31.1 | 34.8 | 22.7 | 30.3 | 12.3 |

[a]The GenBank accession numbers of the viruses listed here are detailed in Tables S4 and S5 in the supplemental material.

(MdPVs), which have only L and S segments, were grouped into three clusters (Group I to III) on the phylogenetic tree (15). OKTV (QDH01 to QDH06, QDR01, QDR02, and QDR04) was assigned to cluster GIII, and pairwise similarity analysis based on the full sequence showed up to 96% nucleotide identity among the currently sequenced OKTVs (Fig. 4F).

*(b) HPTV1 of the Nairoviridae.* Huangpi tick virus 1 (QD-H02) and Henan tick virus (QD-H02) of *Nairoviridae* were also found in *H. flava* in this study. The genomes of the *Nairoviridae* consist of three RNA segments, the S segment encoding the nucleoprotein (N), the M segment encoding glycoprotein (G), and the L segment encoding the RdRp (Fig. 5A and B). HPTV1 QDH02 showed nucleotide identities of 98.32%, 98.79%, and 99.01% with the L (GenBank accession number MW721865.1), M (accession number MW721866.1), and S (accession number MW721868.1) segments of HPTV1 and amino acid identities of 99.57%, 98.31%, and 99.89% with the RdRp (accession number AJG39237.1), G (accession number AJG39276.1), and N (accession number AJG39305.1) proteins of HPTV1, respectively. The HNTV QD-H02 strain showed nucleotide identities of 98.33%, 94.53%, and 92.66% with the L, M, and S segments of HNTV (GenBank accession numbers MZ244224.1 to MZ244226.1) and amino acid identities of 99.07%, 98.08%, and 99.89% with the RdRp, GPC, and N proteins of the HNTV reference strain (accession numbers QYW06749.1 to QYW06751.1), respectively. Both previously identified viruses clustered with a previously reported identical virus species in the phylogenetic tree (Fig. 5C). The two viruses clustered to form a subgroup with Tamdy virus (25) and Tacheng virus 1 (17) of *Orthonairovirus* on the phylogenetic tree (Fig. 5C).

*(c) CPTV3 of Chuviridae.* In this study, five strains of Changping tick virus 3 (QDR01 to QDR04 and QDH03) of *Chuviridae* were identified in *R. sanguineus* from hedgehogs and *D. sinicus* from hares. The virus genome encodes the RdRp, the glycoprotein (G), and the N protein (Fig. 6A). A phylogenetic tree based on the viral RdRp sequences of *Chuviridae* was constructed. CPTV3 strains QDR01 to QDR04 and QDH03 form a subgroup with CPTV3 (GenBank accession number YP_009177707.1). CPTV3 strains QDR01 to QDR04 and QDH03 were clustered with many other viruses isolated from ticks, including Tacheng tick virus 5 (17) and Wuhan tick virus 2 (17) (Fig. 6B). In addition, the

**FIG 4** Legend (Continued)

Okutama tick virus strain QDH01. The viral genome contains two or three segments encoding the RdRp, glycoprotein (G), nucleoprotein (N), or Ns protein. The encoded proteins are marked by differently colored modules. All proteins are drawn to scale for genomic size. (B to F) Phylogenetic trees of *Phenuiviridae* based on the amino acid sequences of the RdRp proteins. The heatmaps of the paired genetic distances are based on the full-length nucleotide sequences. Viruses obtained in ticks here are highlighted in red. The bars indicate nucleotide substitutions per site. The GenBank accession numbers of the viruses listed here are detailed in Table S4 in the supplemental material.

**TABLE 3** Nucleotide and amino acid sequence identities of QDTUV H01 with other *Phenuiviridae*[a]

| Cluster type and virus | % identity with QDTUV H01 | | | | | | |
| --- | --- | --- | --- | --- | --- | --- | --- |
| | Nucleotide | | | Amino acid | | | |
| | L | M | S | RdRp | G | N | Ns |
| **Uukuvirus** | | | | | | | |
| Toyo virus | 77.1 | 71.5 | 45.8 | 90.5 | 81.4 | 85.1 | 73.0 |
| Uukuniemi virus | 54.1 | 46.9 | 32.6 | 48.9 | 36.5 | 35.9 | 18.0 |
| Kaisodi virus | 64.3 | 61.8 | 32.0 | 69.7 | 63.3 | 49.6 | 49.0 |
| Huangpi tick virus 2 | 65.7 | 60.5 | 31.6 | 69.5 | 62.1 | 50.4 | 49.4 |
| Silverwater virus | 65.2 | 61.0 | 32.8 | 70.2 | 63.6 | 50.6 | 44.4 |
| **Bandavirus** | | | | | | | |
| SFTSV | 46.5 | 38.1 | 29.6 | 33.5 | 22.1 | 29.5 | 18.1 |
| **Phlebovirus** | | | | | | | |
| Rift Valley fever virus | 46.8 | 37.5 | 32.6 | 35.8 | 23.4 | 40.9 | 16.3 |

[a]The GenBank accession numbers of the viruses listed here are detailed in Tables S4 and S5 in the supplemental material.

sequence similarity between these five strains and the CPTV3 reference strain was up to 97% (Fig. 6B).

## DISCUSSION

This study found 36 strains of 10 viruses from 4 viral families in *H. flava*, *R. sanguineus*, *D. sinicus*, *H. longicornis*, and *H. campanulata* from hares and hedgehogs in Qingdao, China, by metatranscriptomics. Among them, three novel viruses, QDTIFV, QDTPV, and QDTUV, were observed. Of the seven known viruses, both HfIFV and OKTV were first detected in China. In addition, we also found some short fragments of suspected new viruses, between 2,000 and 3,000 bp in length. They have the highest identities with Xinjiang tick-associated virus 1 and *Ixodes scapularis*-associated virus 2,

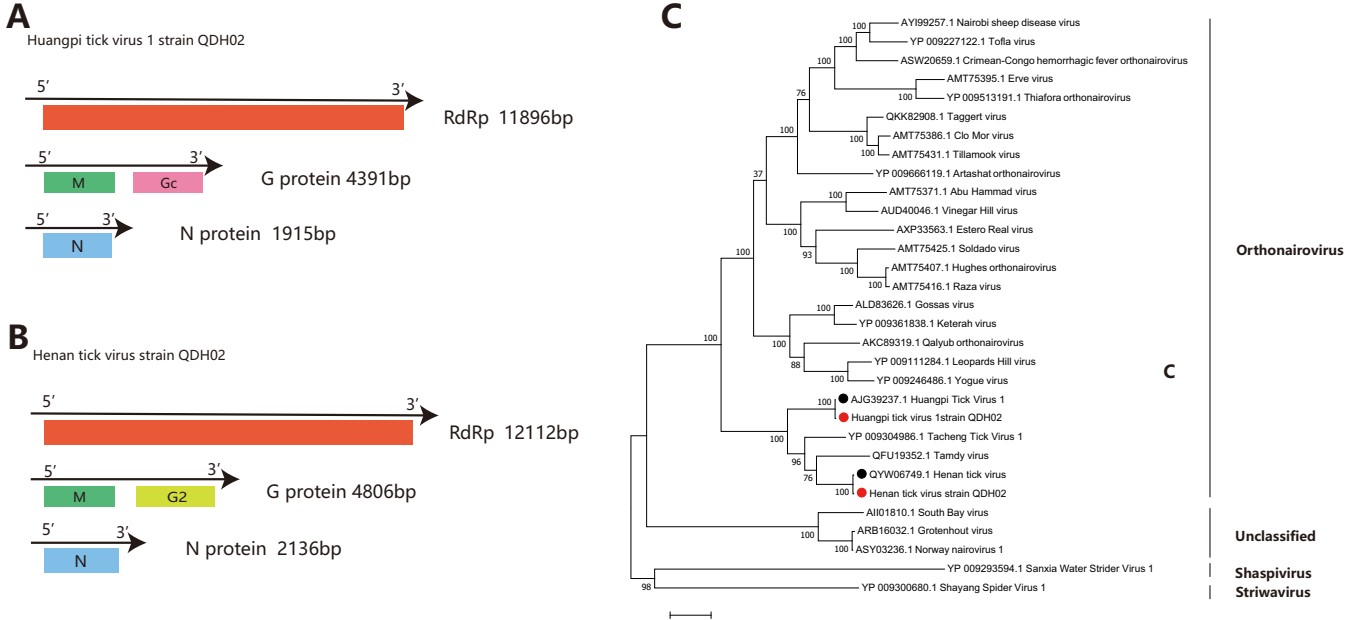

**FIG 5** Genome structure and phylogenetic analysis of *Nairoviridae*. (A and B) Genome structures and putative coding regions of Huangpi tick virus 1 and Henan tick virus. The viral genome contains three segments encoding RdRp, glycoprotein (G), and nucleoprotein (N). The encoded proteins are marked by differently colored modules. All proteins are drawn to scale for the protein size. (C) Phylogenetic analyses of *Nairoviridae*. Phylogenetic trees were constructed based on the RdRp protein sequences of representative viruses of the *Nairoviridae*. Viruses obtained in this study are highlighted in red. The bar indicates nucleotide substitutions per site. The GenBank accession numbers of the viruses listed here are detailed in Table S4 in the supplemental material.

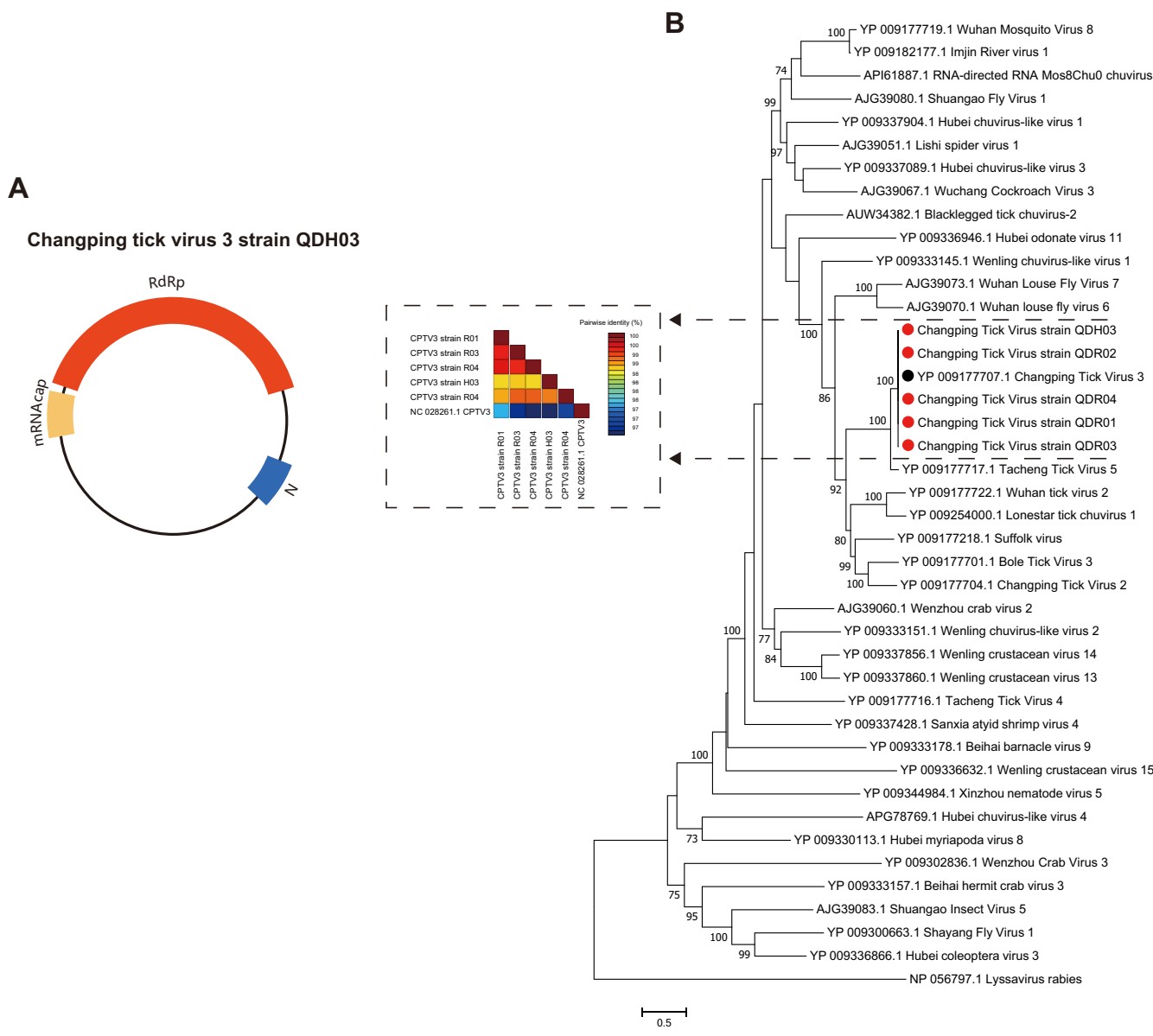

**FIG 6** Genome structure and phylogenetic analysis of *Chuviridae*. (A) Genome structure and putative coding regions of Changping tick virus 3. The viral genome contains three segments encoding RdRp, glycoprotein (G), and nucleoprotein (N). The three proteins are marked by differently colored modules. The three proteins are drawn to scale for genomic size. (B) Phylogenetic tree of *Chuviridae* based on the amino acid sequences of the RdRp domain and heatmap of the paired genetic distances based on the full-length nucleotide sequences. Viruses obtained in ticks here are highlighted in red. The bar indicates nucleotide substitutions per site. The GenBank accession numbers of the viruses listed here are detailed in Table S4 in the supplemental material.

respectively (see Table S3 in the supplemental material). Our results showed that there were large numbers of viruses of the *Phenuiviridae*, *Iflaviridae*, *Nairoviridae*, and *Chuviridae* in the five species of ticks in Qingdao. Our research revealed the diversity and abundance of viruses carried by different tick species (26, 27), which enriches the viral diversity of ticks in China.

*Phenuiviridae* can infect animals, plants, and fungi, which is rare among known virus families (28, 29). *Phenuiviridae* are arboviruses that can replicate in different hosts such as insects, humans, and rice (22, 30). Many kinds of *Phenuiviridae*, such as Dabie banda-virus (3), Rift Valley fever virus (RVFV) (31), Heartland virus (HRTV) (32), and rice stripe virus (33), are highly pathogenic to humans, animals, or plants, and they impose a heavy global burden on human health, animal husbandry and agriculture. SFTSV was first reported in 2011 as a representative virus of the *Phenuiviridae* (3). In recent years, more than 1,000 cases of SFTSV infection have been diagnosed annually in China (34).

Additionally, South Korea, Japan, and Vietnam have also found human cases of SFTSV infection (35–38). Currently, many cases of SFTSV infection and studies on ticks carrying SFTSV have been reported in Shandong, China (39), which suggests that SFTSV is widespread in ticks in Shandong, China. Two newly discovered viruses, QDTPV and QDTUV, also belong to the *Phenuiviridae*. Currently, it is unclear whether the two novel viruses can infect humans or animals to cause new infectious diseases.

Many of the known viruses found in this study were first reported in Japan. These findings indicate that the viruses in China and Japan may originate from the same ancestor. Tick-borne viruses are transmitted along with migratory birds across the East China Sea and/or the Sea of Japan. Globally, migratory birds are transporters of tick-borne pathogens such as *Borrelia*, *Rickettsia*, and Crimean-Congo hemorrhagic fever virus (CCHFV) (40–43), and tick-borne viruses can be transmitted between humans and livestock through travel and trade. In the future, the detection of these viruses in ticks carried by migratory birds would be powerful evidence. To understand the evolutionary relationships between tick-borne viruses in China and Japan, we performed a phylogenetic analysis on 52 TBVs from China and/or Japan in recent years. The results showed that these viruses were classified into at least nine virus families, including *Phenuiviridae*, *Chuviridae*, *Rhabdoviridae*, *Nyamiviridae*, *Flaviviridae*, *Nairoviridae*, *Spinareoviridae*, *Iflaviridae*, and *Orthomyxoviridae* (Fig. 7). The multiple viruses found in the two countries are distributed on branches of the same virus genus on the phylogenetic tree, indicating a close evolutionary relationship. In our study, HfIFV and OKTV strains were found in China, with up to 99% amino acid identity with strains previously found in Japan. QDTPV and QDTUV were also found in China and showed 80.9% and 90.5% amino acid identities of RdRp with Mukawa virus and Toyo virus previously reported in Japan, respectively. The RdRp protein has the highest level of homology compared to other proteins, indicating that RdRp, as an important enzyme in the process of viral replication that participates in the RNA genome replication process, is very evolutionarily conserved. The diversity of the M segment and S protein enhances the adaptability and immune escape of the virus in the host.

Since only the types of viruses carried by ticks on hares and hedgehogs have been identified, the viruses found in this study are not all species of viruses carried by ticks in Qingdao.

In conclusion, 36 strains of 10 viruses from 4 viral families were found using metatranscriptomics sequencing in this study. Three novel TBVs, QDTIFV, QDTPV, and QDTUV, were found in Qingdao, China. Half of the TBVs in this study are genetically closely related to viruses of the same genus from Japan. These findings provide indirect evidence for the cross-sea transmission of TBVs between China and Japan. Therefore, it is clear that there is a great diversity of viruses in blood-sucking insects of wild animals in China.

## MATERIALS AND METHODS

**Sample collection.** A total of 662 ticks were collected from June to July 2019 in the suburbs of Qingdao, Shandong Province, China, of which 420 were collected from hedgehogs and 242 were collected from hares. Ticks were identified by morphological identification and mitochondrial 16S rRNA sequencing (44). All samples were stored at −80°C throughout the whole process.

**RNA library construction and sequencing.** The collected ticks were divided into 10 groups according to tick species, and the number of ticks in each group was between 60 and 80. Next, the ticks in each group were washed with phosphate-buffered saline (PBS) three times. One milliliter of Dulbecco's modified Eagle's medium (DMEM) containing 10% fetal bovine serum (FBS) was added to the ticks in each group, and the ticks were ground at a low temperature to homogenize them completely. Pure RNA was obtained by phenol-chloroform extraction and isopropyl alcohol precipitation. The library of RNA was constructed by using the New England BioLabs (NEB) total RNA library building kit according to the manufacturer's instructions. Paired-end (100 bp) sequencing of each RNA library was performed on the Illumina NovaSeq6000 instrument. The sequencing of the qualified library was performed using an Illumina NovaSeq6000 instrument in PE 2× 150-bp mode.

**Metagenomic bioinformatics analyses.** Quality control of the next-generation sequencing raw data was performed by using fastp v0.20.0 (45), including adapter trimming and quality filtering. Ribosome fragment elimination was estimated by classifying the reads with SortMeRNA v2.0 (version 4.3.2) (46) against the SILVAdb128 small-subunit (SSU) and large-subunit (LSU) databases (47). Sequencing reads aligning to the

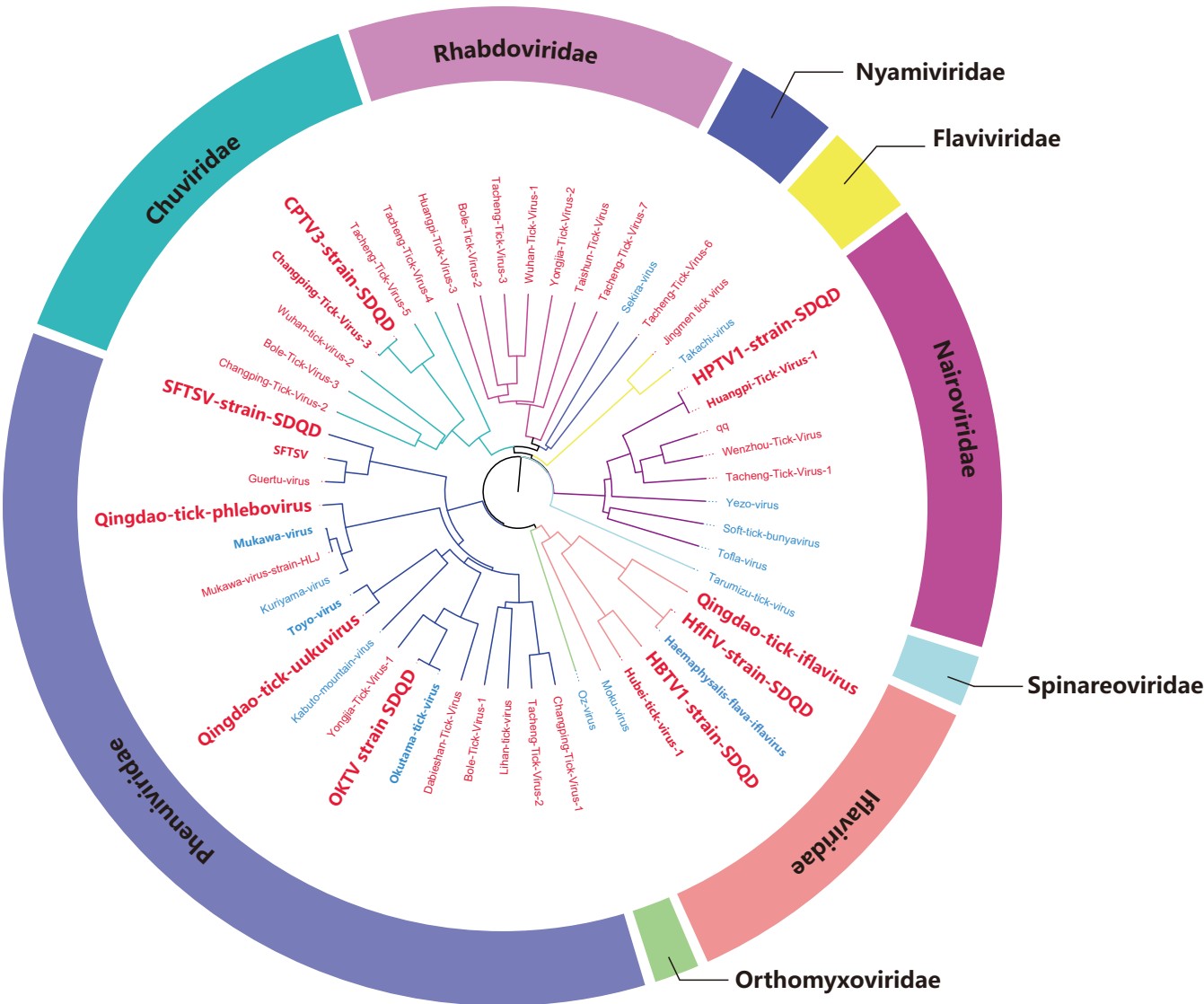

**FIG 7** Phylogenetic analyses of tick-borne viruses from China and Japan. Phylogenetic trees were constructed based on the RdRp sequences. Viruses found in China are shown in red, while viruses found in Japan are shown in blue. The viruses found in this study are shown in bold red.

host reference genomes (*Dermacentor silvarum* [GenBank assembly accession number GCF_013339745.1], *Rhipicephalus annulatus* [mites and ticks] [accession number GCA_013436015.1], *Rhipicephalus sanguineus* [brown dog tick] [accession number GCF_013339695.1], and *Rhipicephalus microplus* [southern cattle tick] [accession number GCF_013339725.1]) by bbmap (version 38.18) (48) were removed. Next, the cleaned sequence reads were obtained as the starting point for later analysis.

The reads were then *de novo* assembled with metaSPAdes v3.13.1. The assembled contigs were classified into known viral orders and families mapping to the NCBI nt database by BLASTn (parameter of an E value of <1e−5). Predicted viral contigs were mapped against sequences in the NCBI nr database by using diamond software (parameter of an E value of <1e−5) (49). The reads were classified into known viral species using kraken2 (50). At the same time, the species of reads were annotated using the NCBI virus database through the diamond BLASTx program (parameter of an E value of <1e−5).

**Validation and annotation of virus genomes.** To rule out false-positive results during assembly, specific PCR primers and probes designed according to viral RdRp or polyprotein sequences were used to verify the virus found by NGS in the original sample using reverse transcription-PCR (RT-PCR) (see Table S2 in the supplemental material for details of detection). The potential open reading frames (ORFs) of virus sequences were predicted using ORF finder (https://www.ncbi.nlm.nih.gov/orffinder/). The conserved regions in the sequence were annotated with a CDD search (https://www.ncbi.nlm.nih.gov/Structure/cdd/wrpsb.cgi).

**Virus classification.** The discovered viruses were classified based on nucleotide and amino acid identities (Table S3). If the nucleotide identity of a new virus species with the whole genome of the reference

virus strain is less than 80% or the amino acid identity with the RNA-dependent RNA polymerase (RdRp) domain of a known virus is less than 90%, the virus is defined as a new virus (27). All new viruses were named "Qingdao tick" (QDT), followed by the names of common viruses classified by them in this study. To distinguish them from other known virus strains, the strains detected in this study are named "QD."

**Phylogenetic analysis.** The nucleotide and amino acid sequences of the viruses from this study were aligned with published reference sequences for the same genus. The nucleotide and amino acid sequences of the alignments were analyzed and compared with representative sequences from GenBank using MAFFT v7.450 (51). Phylogenetic analysis was performed using the full-length nucleotide sequence or the RdRp sequence of the representative members of related viral species or genera. Phylogenetic trees were constructed by the maximum likelihood method based on 1,000 repeated boot-strap replicates using MEGA7 (52). The GenBank accession numbers of the viruses used in this article are detailed in Tables S4 and S5.

**Data availability.** The nucleotide sequence accession numbers of the viruses detected in this study have been submitted to the international nucleotide sequence database (GenBank) and the National Microbiology Data Center (NMDC) (Table S3). All sequence reads generated in this project are available in the NCBI Sequence Read Archive under BioProject accession number PRJNA935925. All viral genome sequences have been submitted to the GenBank database under accession numbers OQ513629 to OQ513687 (Table S3).

## SUPPLEMENTAL MATERIAL

Supplemental material is available online only.
**SUPPLEMENTAL FILE 1**, PDF file, 7.6 MB.

## ACKNOWLEDGMENTS

We thank the Qingdao Center for Disease Control and Prevention for its contribution to the collection and transportation of ticks. We thank the National Microbiology Data Center for its assistance with data uploading.

This work was supported by the Development Fund Project of the State Key Laboratory for Infectious Disease Prevention and Control under grant 2011SKLID104, the National Resource Bank under grant NPRC-32, the Research and Application Demonstration of Key Supporting Technologies of Pathogenic Microorganism Resource Bank under grant 2022YFC2602202, and the Technical System and Standard for Comprehensive Response to Human Virus Infection under grant 2022YFC2602402.

We have no conflicts of interest to disclose.

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
