## [Reviewer comments · Microbiology Spectrum]

Microbiology Spectrum

Diversity analysis of tick-borne viruses from hedgehogs and hares in Qingdao, China

Jun Han, Geng Hu, Fachun Jiang, Qin Luo, Kexin Zong, Liyan Dong, Guoyong Mei, Haijun Du, Hongming Dong, Qinqin Song, Juan Song, Zhiqiang Xia, and Chen Gao

Corresponding Author(s): Jun Han, Chinese Center for Disease Control and Prevention

Review Timeline:

Submission Date:	January 3, 2023
Editorial Decision:	February 9, 2023
Revision Received:	March 3, 2023
Accepted:	March 12, 2023

Editor: Jonathan Jacobs

Reviewer(s): Disclosure of reviewer identity is with reference to reviewer comments included in decision letter(s). The following individuals involved in review of your submission have agreed to reveal their identity: Li Guo (Reviewer #1); Tao Shen (Reviewer #2)

Transaction Report:

DOI: <https://doi.org/10.1128/spectrum.05340-22>

February 9, 2023

Dr. Geng Hu
Chinese Center for Disease Control and Prevention
155 Changbai Rd, Beijing 102206, China
Beijing
China

Re: Spectrum05340-22 (Diversity analysis of tick-borne viruses from Hedgehogs and Hares in Qingdao, China)

Dear Dr. Geng Hu:

Thank you for your submission, Diversity analysis of tick-borne viruses from Hedgehogs and Hares in Qingdao, China, to ASM Microbiology Spectrum. I have reviewed the manuscript, and have received two independent reviews as well. As it stands, the manuscript will require substantial revisions prior to publication in 'Spectrum'. The requested revisions are listed below:

1. I encourage you to work with an English technical language editor to improve the quality of the writing, correct several grammatical errors, and correctly apply scientific notation and formatting where applicable.
2. Increase the font size for labels used in your figures. As it stands, many of the labels are unreadable in printed format.
3. Revise your bioinformatics methods section. Spelling and grammar errors were substantial. No information was provided on the depth of sequencing or results pre- and post-QC of the raw sequencing reads - this should be corrected. MetaSpades is used for de novo assembly, and reads are not "spliced". Clarify why was a 90% cut-off used for the breakpoint for new species delineation (is there a reference)?
4. Please indicate whether the assemblies submitted to GenBank and NMDC is described as a Metagenome Assembled Genome (MAG) or not (it should be).
5. Please address all the concerns of Reviewer 1 and 2 as well.

Link Not Available

Sincerely,

Jonathan Jacobs

Journals Department
Reviewer comments:

Reviewer #1 (Comments for the Author):

Some animals can carry viruses and come in close contact with humans, which creates opportunities to transmit virus to humans. Hu et al. found 36 strains of ten viruses from four viral families from hares and hedgehogs by metatranscriptomics in Qindao, China. Among them, three novel viruses, QDTIFV, QDTPV, and QDTUV were observed. Of the seven known viruses, both HflFV and OKTV were first detected in China. These findings enrich the viral spectrum of ticks. It is important for surveillance the emerging infectious disease.

However, there are several points to concern:

1. Line 150-152: "The pairwise similarity analysis based on full sequence showed up to 94% nucleotide identity between the currently sequenced HBTUV1(QD-H02-04, R01, R03, R04) (Figure 3C)." However, lines 164-165 in the Figure legend of Figure 3 (C): "Phylogenetic tree and pairwise genetic distance heatmap of iflaviruses, based on RdRp domain amino acid sequences." Please keep in consistent with "full sequence, nucleotide acid identity" or "RdRp domain amino acid sequences".
2. Lines 209-210: "the pairwise similarity analysis based on full sequence showed up to 90% nucleotide acid identity between four QDTUV strains (H01, H04-06) (Figure 4F)." However, lines 221-222: in the Figure legend of Figure 4 B-4F): "Phylogenetic tree and 222 pairwise genetic distance heatmap of Phenuiviridae, based on RdRp domain amino acid sequences". Please keep in consistent with "full sequence, nucleotide acid identity" or "RdRp domain amino acid sequences".
3. For these first detected HflFV and OKTV, whether the authors isolated virus successfully? It is important for the follow-up work.
4. The English need to polish in standard English.

Reviewer #2 (Comments for the Author):

In this paper, metagenomic technology was used to construct a microbiogram library carried by different ticks in Qingdao, Shandong Province, China. *Haemaphysalis flava*, *Rhipicephalus sanguineus*, *Dermacentor sinicus*, *Haemaphysalis longicornis*, *Haemaphysalis campanulate* were collected to study their microbial classification, and all viruses and bacteria in tick samples were identified by metagenomics to build a tick microbial resource bank. The data of different tick microbiome were systematically analyzed at the taxonomic level, and the genetic evolution of important tick-borne viruses was analyzed. It was found that the abundance of bacteria and viruses varied greatly among different ticks, which made further comparative studies of microbial communities carried by ticks possible.

1. Pay attention to the wording of the article. The wording of concluding statements should be more careful, and pay attention to the fluency and consistency of the full text language;
2. The English name of the species shall be italicized.
3. Severe fever with thrombocytopenia syndrome virus was mentioned in the method part of the abstract, while Dabie Banda virus was mentioned in the following articles, which should be consistent.
4. The conclusion part of the abstract only includes the microbiome library, and the conclusion of Dabie Banda virus evolution analysis should be added.

Staff Comments:

Preparing Revision Guidelines

- Point-by-point responses to the issues raised by the reviewers in a file named "Response to Reviewers," NOT IN YOUR COVER LETTER.
- Upload a compare copy of the manuscript (without figures) as a "Marked-Up Manuscript" file.
- Each figure must be uploaded as a separate file, and any multipanel figures must be assembled into one file.
- Manuscript: A .DOC version of the revised manuscript

- Figures: Editable, high-resolution, individual figure files are required at revision, TIFF or EPS files are preferred

Please return the manuscript within 60 days; if you cannot complete the modification within this time period, please contact me. If you do not wish to modify the manuscript and prefer to submit it to another journal, please notify me of your decision immediately so that the manuscript may be formally withdrawn from consideration by Microbiology Spectrum.

In this paper, metagenomic technology was used to construct a microbiogram library carried by different ticks in Qingdao, Shandong Province, China. *Haemaphysalis flava*, *Rhipicephalus sanguineus*, *Dermacentor sinicus*, *Haemaphysalis longicornis*, *Haemaphysalis campanulate* were collected to study their microbial classification, and all viruses and bacteria in tick samples were identified by metagenomics to build a tick microbial resource bank. The data of different tick microbiome were systematically analyzed at the taxonomic level, and the genetic evolution of important tick-borne viruses was analyzed. It was found that the abundance of bacteria and viruses varied greatly among different ticks, which made further comparative studies of microbial communities carried by ticks possible.

1. Pay attention to the wording of the article. The wording of concluding statements should be more careful, and pay attention to the fluency and consistency of the full text language;
2. The English name of the species shall be italicized.
3. Severe fever with thrombocytopenia syndrome virus was mentioned in the method part of the abstract, while Dabie Banda virus was mentioned in the following articles, which should be consistent.
4. The conclusion part of the abstract only includes the microbiome library, and the conclusion of Dabie Banda virus evolution analysis should be added.

Response to Reviewers

Microbiology Spectrum

MS No.:Spectrum05340-22R1

MS Title: Diversity analysis of tick-borne viruses from hedgehogs and hares in Qingdao, China

Response to Jonathan Jacobs Editor:

I am very grateful to your comments for the manuscript. According to your advice, we amended the relevant part in manuscript. Your questions were answered below.

1. I encourage you to work with an English technical language editor to improve the quality of the writing, correct several grammatical errors, and correctly apply scientific notation and formatting where applicable.

RE: Thank you for your comments. In view of the language problems you mentioned, we revised some grammar and vocabulary errors in the manuscript. And we have marked the modifications with red annotations.

2. Increase the font size for labels used in your figures. As it stands, many of the labels are unreadable in printed format.

RE: Thank you for your advice. To facilitate the label on the figures to be read in printed format, we have increased the font size in the figures of revised manuscript. We adapted the design of Figure 4 to make it more suitable for the printing format. For the other pictures, we have adjusted the font size and uploaded the EPS format pictures for your viewing.

3. Revise your bioinformatics methods section. Spelling and grammar errors were substantial. No information was provided on the depth of sequencing or results pre- and post-QC of the raw sequencing reads - this should be corrected. MetaSpades is used for de novo assembly, and reads are not "spliced". Clarify why was a 90% cut-off used for the breakpoint for new species delineation (is there a reference)?

RE: We improved the description of the bioinformatics using professional terms in the methods section. Spelling and grammar errors of the whole manuscript were corrected.

We previously showed some information of sequencing or results pre- and post-QC of the raw sequencing reads in the Table 1 of supplementary material 3, which may not be detailed enough, so we added some information to introduce. In addition, the novel viruses were named according to a reference (Guo, L. *et al.* Virome of Rhipicephalus ticks by metagenomic analysis in Guangdong, southern China. *Front Microbiol* **13**, 966735, doi:10.3389/fmicb.2022.966735 (2022)). The reference was cited in the *Virus classification* of Methods of revised manuscript.

4. Please indicate whether the assemblies submitted to GenBank and NMDC is described as a Metagenome Assembled Genome (MAG) or not (it should be).

RE: The data was uploaded is the Metagenome Assembled Genome. All sequence reads generated in this project are available in the NCBI Short Read Archive under BioProject PRJNA935925.

5. Please address all the concerns of Reviewer 1 and 2 as well.

RE: We carefully read the reviewer's comments and revised the manuscript according to two reviewers' requirements. The responses are given below.

Response to Reviewer 1:

1. Line 150-152: *"The pairwise similarity analysis based on full sequence showed up to 94% nucleotide identity between the currently sequenced HBTV1(QD-H02-04, R01, R03, R04) (Figure 3C)." However, lines 164-165 in the Figure legend of Figure 3 (C): "Phylogenetic tree and pairwise genetic distance heatmap of iflaviruses, based on RdRp domain amino acid sequences." Please keep in consistent with "full sequence, nucleotide acid identity" or "RdRp domain amino acid sequences".*

RE: We are sorry for our carelessness. We modified the associated contents in the revised manuscript. The phylogenetic tree analysis is based on the RdRp sequence of the virus, while the homology analysis is based on the full nucleotide length of the virus.

2. Lines 209-210: *"the pairwise similarity analysis based on full sequence showed up to 90% nucleotide acid identity between four QDTUV strains (H01, H04-06) (Figure 4F)." However, lines 221-222: in the Figure legend of Figure 4 B-4F): "Phylogenetic tree and 222 pairwise genetic distance heatmap of Phenuiviridae, based on RdRp domain amino acid sequences". Please keep in consistent with "full sequence, nucleotide acid identity" or "RdRp domain amino acid sequences".*

RE: Reply as above. And we corrected a similar mistake in the note of Figure 6..

3. *For these first detected HfIFV and OKTV, whether the authors isolated virus successfully? It is important for the follow-up work.*

RE: We found the whole sequence of three new viruses through NGS, but we have not yet isolated the new viruses, but we are still working on it.

4. *The English need to polish in standard English.*

RE: We have revised the errors and improved the sentence of the whole manuscript.

Response to Reviewer 2:

1. Pay attention to the wording of the article. The wording of concluding statements should be more careful, and pay attention to the fluency and consistency of the full text language;

RE: Thank you for your comments. We improved the sentence of the whole manuscript as above described.

2. The English name of the species shall be italicized.

RE: We are sorry for our carelessness. We checked the standardization of fonts in the manuscript and corrected some errors

3. Severe fever with thrombocytopenia syndrome virus was mentioned in the method part of the abstract, while Dabie Banda virus was mentioned in the following articles, which should be consistent.

RE: Thank you for your comments. Severe fever with thrombocytopenia syndrome virus (SFTSV) was the name of the virus when it was discovered, and now it was named Dabie Banda virus by ICTV. In order not to cause confusion, the name of the virus in the revised manuscript is Dabie Banda virus. When the virus first appears in the manuscript, SFTS virus is marked in brackets.

4. The conclusion part of the abstract only includes the microbiome library, and the conclusion of Dabie Banda virus evolution analysis should be added.

RE: In our study, 10 viruses with a total of 36 strains were identified. Due to space limitations, we cannot discuss Dabie bandavirus in detail in the abstract, but Dabie bandavirus identified in this study is described in detail in the results and discussion.

March 12, 2023

Dr. Jun Han
Chinese Center for Disease Control and Prevention
Center for Viral Resource
155 Changbai Road
Beijing 102206
China

Re: Spectrum05340-22R1 (Diversity analysis of tick-borne viruses from hedgehogs and hares in Qingdao, China)

Dear Dr. Jun Han:

thank you for your revised manuscript and the extensive improvements you have made.

Your manuscript has been accepted, and I am forwarding it to the ASM Journals Department for publication. You will be notified when your proofs are ready to be viewed.

Sincerely,

Jonathan Jacobs
Editor, Microbiology Spectrum
